# Lightweight Multiprincipal Element Alloys with Excellent Mechanical Properties at Room and Cryogenic Temperatures

**DOI:** 10.3390/e24121777

**Published:** 2022-12-05

**Authors:** Gongxi Lin, Ruipeng Guo, Xiaohui Shi, Lina Han, Junwei Qiao

**Affiliations:** 1College of Materials Science and Engineering, Taiyuan University of Technology, Taiyuan 030024, China; 2Key Laboratory of Interface Science and Engineering in Advanced Materials, Ministry of Education, Taiyuan University of Technology, Taiyuan 030024, China

**Keywords:** lightweight multiprincipal element alloys, microstructure, mechanical properties, strengthening mechanism

## Abstract

Lightweight multiprincipal element alloys (MPEAs) are promising candidates for potential application as engineering materials due to their high strength and low density. In this work, lightweight Ti_70_Al_15_V_15_ and Ti_80_Al_10_V_10_ MPEAs were fabricated via vacuum arc melting. The phases of the Ti_70_Al_15_V_15_ alloys consisted of a BCC phase and a small amount of B2 phase while the Ti_80_Al_10_V_10_ alloys displayed a dual-phase structure with BCC and HCP phases. The different phase compositions led to differences in their mechanical properties. When the temperature changed from 298 K to 77 K, the strength of the alloys further increased and maintained a certain plasticity. This is attributed to the increasing lattice friction stress at cryogenic temperature. TEM observation demonstrated that dislocation played a crucial role in plastic deformation for both the Ti_70_Al_15_V_15_ and Ti_80_Al_10_V_10_ alloys. In addition, Ti_80_Al_10_V_10_ exhibited significant work-hardening capabilities. By analyzing the strengthening mechanism of the alloys, the theoretical yield strength was calculated, and the results agreed with the experimental values. The present results provide new insight into developing lightweight MPEAs containing Ti and Al.

## 1. Introduction

An emerging class of alloys termed “multiprincipal element alloys” (MPEAs) or “high entropy alloys” (HEAs) which contain significant atomic fractions of several elements have aroused considerable research attention as potentially high-strength and ductile material [1,2]. MPEAs usually obtain their properties from multiple base elements rather than the main components, which offers the possibility to explore excellent performances in a wide composition space. It has been reported that body-centered-cubic (BCC) MPEAs generally exhibit high strength but limited plasticity, while face-centered-cubic (FCC) MEPAs have relatively low strength but high plasticity [3]. For example, the CoCrFeMnNi HEA with an FCC structure displays a remarkable tensile ductility of 60%, but the yield strength is only 197 MPa. The classic BCC refractory TiZrHfNbTa HEA has a high yield strength of 1195 MPa but a low tensile ductility of 4.7% [4,5]. In addition, these MPEAs mainly contain relatively heavy elements, e.g., Co, Cr, Fe, Mn, and Ni for FCC structures and Hf, Nb, Ta, Mo, and W for BCC structures [1,6]. The high density of MPEAs ranging from 7 to 24 g/cm^3^ [7,8,9] seriously limits their processing and applications in the aerospace, automotive, and electronics industries. Therefore, it is still a challenge to develop new lightweight MPEAs with excellent mechanical properties.

Many light elements, such as Li, Mg, Al, Sc, Ti, and V, have been used to design lightweight MPEAs to reduce their density. Juan et al. developed the Al_20_Be_20_Fe_10_Si_15_Ti_35_ low-density MPEA, whose microstructure contains one main phase and two minor phases, and its density is 3.91 g/cm^3^ and the hardness is 911 HV [10]. Youssef et al. prepared Al_20_Li_20_Mg_10_Sc_20_Ti_30_ MPEAs with a density of 2.67 g/cm^3^ using mechanical alloying, which has a single-phase FCC structure [11]. The great differences in the light-element atoms lead to a large lattice distortion and solid solution strengthening, and most lightweight MPEAs exhibit high strength and hardness. Meanwhile, more brittle intermetallic phases may be formed in lightweight MPEAs, such as B2 and laves phases, and the tensile ductility is generally poor [12,13]. Developing lightweight MPEAs with both high strength and ductility is critical and challenging work.

In this work, lightweight Al, Ti, and V elements were used to develop new lightweight MPEAs due to their similar atomic size and electronegativity as well as chemical stability. Recently, some of the lightweight HEAs consisting of light elements such as Al and Ti have been investigated [14,15]. Al and Ti tend to form stable intermetallic compounds with many other elements due to their high negative values of heat of mixing [16]. These intermetallic compounds have always deteriorated the mechanical properties of lightweight HEAs since a higher concentration of Al atoms often results in an ordered BCC crystal structure (B2), and the ordered phase usually exhibits embrittlement at ambient temperature [17]. When the content of Ti is increased, the degree of ordering in the AlNbTiZr alloy can be reduced, resulting in a certain plasticity [18]. Thus, lightweight nonequiatomic Ti_70_Al_15_V_15_ and Ti_80_Al_10_V_10_ MPEAs are designed and synthesized based on equiatomic TiAlV alloy [19]. By adjusting the content of Ti and Al, the present MPEA achieves a low density with well-balanced strength–ductility synergy. Links between the chemical composition, phase constitution, hardness, and mechanical properties are well established.

## 2. Materials and Methods

Two kinds of lightweight MPEAs with nominal compositions of Ti_70_Al_15_V_15_ and Ti_80_Al_10_V_10_ (atomic percent, at. %) were prepared by arc melting a mixture of the constituent elements with purity higher than 99.9% under a Ti-getter argon atmosphere. These alloys were denoted as Ti70 and Ti80, respectively. To achieve a homogeneous distribution of elements, the melting and solidification processes were repeated at least five times. Then, the 80 × 10 × 6 mm^3^ (length × width × thickness) thin plates were prepared by a water-cooled copper mold suction casting.

Rectangular dog-bone tensile specimens with a gauge length of 10 mm and cross-section dimension of 3 × 1.5 mm^2^ were prepared by electrical discharge machining and mechanically polished to 2000 grit size. An Instron 5969 universal testing machine was used to carry out uniaxial tensile tests at a constant strain rate of 1 × 10^−3^ s^−1^ at ambient temperature (298 K) and liquid nitrogen temperature (77 K). There were at least three samples used to ensure the reliability of the experimental data. X-ray diffraction (XRD) was conducted to investigate the crystal structure of the alloys in a 2θ (diffraction angle) range from 20° to 100° using a PHILIPS APD-10D diffractometer with Cu Kα radiation. Microstructures were characterized by scanning electron microscopy (SEM) under a Phenom XL equipped with energy dispersive spectroscopy (EDS) and by transmission electron microscopy (TEM) with a JEM-F200 electron microscope operated at 200 KV. For SEM analysis, the surfaces of samples were ground to 2000 grit SiC paper and then electrochemically polished using a HClO_4_:CH_3_(CH_2_)_3_OH:CH_3_OH = 1:7:12 solution (in a volume percent, vol %) with a voltage of 17 V at 298 K. Polished surfaces were etched in a solution of 10 mL HF + 10 mL HNO_3_ + 80 mL H_2_O. For TEM observations, the specimens were first mechanically ground to a thickness below 50 μm and then punched into 3 mm diameter discs. Finally, the discs were thinned to a thickness of electron transparency by twin-jet electropolishing at a voltage of 30 V and a temperature of around −25 °C with the same electrolyte. The theoretical density was calculated by the rule of mixture (RoM) while the actual density was measured by Archimedes’ principle.

## 3. Results

### 3.1. Phase Formation

An equiatomic TiAlV lightweight multiprincipal element alloy has been reported which exhibited excellent creep resistance compared to Ti6Al4V alloys at room and high temperatures [19]. However, the compression or tensile mechanical properties have not been conducted, which may be related to the limited sample size due to brittleness. The results of several studies have demonstrated that adjusting the content of Ti and Al can reduce the ordering of the alloys and inhibit the formation of B2 phases [17,18,20]. Therefore, lightweight MPEAs containing a high Ti content have been widely reported. For example, Ti_65_(AlCrNb)_35_ [21], Ti_3.6_Al_0.3_Zr_1.2_V_0.8_ [18], and Ti_3_Zr_1.5_NbVAl_x_ [22] alloys display high tensile strength beyond 750 MPa and ductility of at least 10%. In the present work, Ti70 and Ti80 MPEAs were designed and studied. Many efforts have been made to study MPEAs’ phase formation rules to accelerate the discovery process. It is well-known that many parameters have been used as the criteria for the prediction of the structure of MPEAs, such as the mixing entropy (Δ*S*_mix_), mixing enthalpy (Δ*H*_mix_), comprehensive parameter of Δ*S*_mix_, and Δ*H*_mix_ (Ω), atomic size difference (*δ*), and valence electron concentration (VEC) [23,24,25,26]. These are defined by the equations below:(1)$$\Delta S_{mix}=-R\sum_{1}^{n}C_{i}\ln C_{i}$$
(2)$$\Delta H_{mix}=\overset{n}{\underset{i=1,i\neq j}{\sum }}\Omega_{ij}C_{i}C_{j}$$
(3)$$\Omega =\frac{T_{m}\Delta S_{mix}}{\left|\Delta H_{mix}\right|}$$
(4)$$\delta =100\sqrt{\overset{n}{\underset{i=1}{\sum }}C_{i}{\left(1-\frac{r_{i}}{r_{a}}\right)}^{2}}$$
(5)$$VEC=\sum_{1}^{n}C_{i}{\left(VEC\right)}_{i}$$
(6)$$\Delta \chi =\sqrt{\overset{n}{\underset{i=1}{\sum }}C_{i}{\left(\chi_{i}-\;\bar{\chi }\right)}^{2}}$$ where *n* is the number of components in an alloy system, and $C_{i}$ and $C_{j}$ are the atomic percentages of the *i*th and *j*th elements, respectively. $\Omega_{ij}=4\Delta H_{ij}^{mix}$ is the regular melt-interaction parameter between the *i*th and *j*th elements, and $\Delta H_{ij}^{mix}$ is the mixing enthalpy of binary alloys. The melting temperature of an n-element alloy, *T_m_*, is calculated using the RoM as $T_{m}=\sum_{1}^{n}C_{i}T_{i}$. $r_{i}$ is the atomic radius of the ith element, $r_{a}=\sum_{1}^{n}C_{i}r_{i}$ is the average atomic radius. $\chi_{i}$ is the Pauling electron negativity of the *i*th element, and ${\bar{\chi }}_{i}=\sum_{1}^{n}C_{i}\chi_{i}$ is the average electronegativity. ${\left(VEC\right)}_{i}$ is the VEC for an individual element.

The criteria of $\Delta S_{mix}$, $\Delta H_{mix}$, Ω, δ, and VEC for the formation of solid solutions and their values for Ti70 and Ti80 alloys are listed in Table 1. Yang et al. defined the criteria for the formation of random solid solutions in MPEAs to be in the range of −15 < Δ*H_mix_* < 5 kJ/mol and 1% < δ <6.6% [27]. The Ti70 and Ti80 MPEAs in this study conformed to the specified range. The enthalpy increased significantly with increasing *T_i_* content, indicating that the formation of precipitation or intermetallic compounds in the alloy was effectively suppressed [15]. At the same time, changes in the enthalpy and entropy values led to a larger parameter, Ω, which are beneficial for the formation of solid solutions in MPEAs. In addition, VEC is another key parameter which can be used to determine the crystal type. According to Guo et al. [23], FCC solid solutions are stable with VEC values more than eight whereas BCC solid solutions are generated with VEC values less than 6.87. The VEC values of the three elements were close, and the high *T_i_* content resulted in the same VEC value of four for both MPEAs [24]. The judgment criteria for the correlation between the electronegativity mismatch were given by Dong et al. [28]. Moreover, Δχ < 11.7% favored the formation of a single-phase solid solution. The MPEAs designed in this study meet the empirical parameters required for the formation of solid solutions.

### 3.2. Density, XRD, and Microstructure

Table 2 shows the theoretical densities calculated based on the RoM and the measured densities using the Archimedes method. The measured densities approached their theoretical values, and the density calculation formula is expressed as follows [8]:(7)$$\rho =\frac{\sum C_{i}A_{i}}{\sum \frac{C_{i}A_{i}}{\rho_{i}}}$$ where $C_{i}$, $A_{i}$, and $\rho_{i}$ represent the atomic percentage, atomic weight, and density of the *i*th constituent element, respectively. Figure 1 illustrates the XRD patterns of the as-cast Ti_70_Al_15_V_15_ and Ti_80_Al_10_V_10_ alloys. It is observed that the Ti70 alloys had one set of sharp diffraction peaks that belong to a BCC crystal structure. The Ti80 alloys exhibited BCC and hexagonal close-packed (HCP) crystal structures.

**Table 2 entropy-24-01777-t002:** The measured and calculated of alloys.

Nominal Composition (at.%)	Measured Density (g/cm^3^)	Theoretical Density (g/cm^3^)
Ti_70_Al_15_V_15_	4.49	4.42
Ti_80_Al_10_V_10_	4.53	4.45

The microstructures of both Ti70 and Ti80 alloys were characterized by SEM observations. As shown in Figure 2, both alloys exhibited an approximately equiaxed feature of grains without significant dendrite features. Figure 2a shows the microstructure of Ti70 alloys, and no other precipitated phases were found in the enlarged view of Figure 2b. The Ti80 alloys contained a large number of lamellar secondary phases within the equiaxed grains in Figure 2c,d. According to XRD analysis, the secondary phase was the HCP phase and, the equiaxed grain had a BCC structure. The average sizes of the equiaxed grains were measured from their fractions by employing ImagePro software. At least five images were used for obtaining the average values. The values of the grain size were calculated as 215 μm and 221 μm for the Ti70 and Ti80 alloys, respectively. The average width and length of the precipitated phases for Ti80 were 1 μm and 5 μm, respectively.

### 3.3. Mechanical Properties

Figure 3 exhibits the mechanical properties of the Ti70 and Ti80 alloys at room and cryogenic temperatures. Typical engineering stress-strain curves are presented in Figure 3a. When decreasing the temperature from 298 K to 77 K, the yield strength (YS) of the Ti70 alloys exhibited a significant increase from 536 MPa to 895 MPa. Meanwhile, the uniform elongation (UE) greatly decreased from 13% to 3%. For the Ti80 alloys, they had a yield strength of 657 MPa and tensile ductility of 10% at 298 K. As the temperature reduced to 77 K, the yield strength increased by 155 MPa, and the ductility remained at 9%. The true stress-strain curves are depicted in Figure 3b, and the Ti70 alloys with single-phase BCC structure had a slight work-hardening capacity before fracturing at room temperature. In contrast, the Ti80 alloys exhibited remarkable work-hardening ability in both room and low temperatures. This may be attributed to differences in crystal structures. Conventional titanium alloys typically exhibit limited strain-hardening capability. Changing the phase composition or stress-induced martensitic transformation and deformation twinning by composition design are the main approaches to improve the work-hardening ability [29]. Ti6Al4V is a relatively mature alloy for commercial applications which has an ultimate tensile strength in the cast state of 800 MPa to 900 MPa and a plasticity between 5% and 13% [30,31]. In comparison, the mechanical properties of Ti_80_Al_10_V_10_ alloys are similar to Ti6Al4V, and further component optimization may be required to improve performance. The deformation mechanism of the alloys in this study will be analyzed in later sections.

Figure 4 shows the specific yield strength and density of the current MPEAs compared with those of reported HEAs and titanium alloys [3,6,18,22,29,32,33,34,35,36,37,38,39,40,41]. The density of the current alloys is lower than that of most reported HEAs and is similar to titanium alloys. The specific strength is defined as the yield strength divided by their density. The maximum specific strength reached 202 (MPa·cm^−3^/g) in this study. It is apparent that the specific yield strength in this study was much higher than the reported refractory HEAs and 3D-transition HEAs. Most refractory HEAs are brittle and have no work-hardening capability. Up to now, very limited alloy systems have been found with obviously macroscope tensile ductility, including TiZrHfNb, TiZrHfNbTa, NbMoTaW, etc. [3,42]. The specific strength of refractory HEAs is relatively low due to the high density of refractory elements despite their high yield strength. 3d-transition HEAs possess high plasticity but low yield strength while their density is higher compared to the present lightweight alloys. Low-density HEAs are based on refractory HEAs or 3d-transition HEAs with the addition of light elements, and their overall alloy density does not decrease much. It is known that the addition of light elements, such as Al and Ti, may lead to the formation of intermetallic compounds. Thus, their contents are relatively small. Obviously, the Ti70 and Ti80 alloys exhibited significant advantages over existing HEAs in terms of specific strength, showing enormous potential for advanced structural applications.

### 3.4. Deformation Mechanisms

To reveal the influences of the microstructures on the deformation mechanisms, the deformation structures of both Ti70 and Ti80 alloys after tension were examined by TEM investigation. Figure 5a,b shows the fine microstructure of Ti70 alloys after tensile deformation at room temperature. A high density of dislocations can be found in the bright field (BF) TEM image in Figure 5a. Dislocation generation and interaction is the key to obtaining better strength and toughness [3]. Figure 5b shows an enlarged view of the yellow rectangular area in Figure 5a. The dislocation walls composed of cell substructures were well-developed in the postdeformed samples. It is an accepted postulate that dislocations agglomerate into cells or subgrains during plastic deformation since this configuration represents minimum strain energy for a given dislocation number. Therefore, there is no doubt that the evolution of the dislocation cell was a type of release of distortion energy, which in turn contributed to the plasticity of the alloys [43]. Dislocation slips can be found in some regions with low dislocation density in Figure 5b. The selected area diffraction (SAED) pattern along the [001] zone axis confirmed a BCC phase structure, and another B2 phase was available. According to the similar composition, the appearance of the B2 phase was related to the high Al content [22]. Figure 5c shows the BF TEM image of Ti70 alloys after plastic deformation at 77 K, and it can be obtained that the dislocation density was lower compared to that at 298 K. According to previous reports, low temperature increases the critical resolved shear stress (CRSS) of dislocation slips and makes dislocation movement difficult, resulting in an increase of the strength of the alloys [44]. In Figure 5d, some curved dislocation lines can be found, which are explained by the dislocation encountering obstacles during the slip process. The SAED pattern in the inset reflects the coexistence of BCC and B2 phases. The B2 phase was an obstacle in the slip path that increased the resistance to dislocation motion [3]. In addition, a parallel distribution of dislocations was available in some matrix regions. According to the analysis of the TEM results, it was concluded that no phase transformation or twinning happened during the plastic deformation for the Ti70 alloys at room and cryogenic temperatures. The Ti70 alloys achieved a synergy of strength and plasticity through the slip and interaction of dislocations during plastic straining.

Figure 6 shows the microstructure of Ti80 alloys subjected to plastic deformation at room and cryogenic temperatures using TEM. Figure 6a,b is a BF TEM image of Ti80 alloys after fracturing at room temperature. In Figure 6a, the phase interface hindered the transmission of slip bands and some accumulation of dislocations can be found. Similar results have been reported in alloys containing Ti [45,46]. Phase interface acts as a strong barrier to reduce the effective slip length of dislocations. In addition, it was observed that there were crossdistributed fine HCP phases within the HCP phases. An enlarged view of the yellow rectangular area shows the dislocation wall and dislocation tangle in Figure 6b. The high-density dislocation structures effectively increased the interaction force between dislocations and significantly reduced the average free path for further dislocation slipping, which resulted in the high ultimate tensile strength of alloys [47]. Due to the high-density distribution of the coarse and fine secondary phases in the matrix, only the HCP structure was indexed from the SAEDpattern, as shown in the inset. According to the TEM results, the phase interface formed an obstacle to dislocation movement, resulting in significant work hardening in the Ti80 alloys. Figure 6c,d displays the TEM image of the Ti80 alloys after deformation at cryogenic temperatures. The phase interface formed a barrier to the dislocations, which is similar to the room deformation behavior in Figure 6c. The dislocation tangle at the phase interface is presented in Figure 6d. The SAEDpattern in the inset confirms that the lamellar structure had an HCP structure. The crossdistributed lamellar HCP phases contacted and interfered with each other during the deformation process, which is conducive to dislocation pile-ups and entanglement, thus maintaining excellent strength and plasticity of the alloys [48].

### 3.5. Strengthening Mechanisms

A systematic understanding of the mechanical properties in this work can be obtained from the aspect of the strengthening model. In general, the strengthening mechanisms of an alloy mainly include grain boundary hardening ($\sigma_{g}$), solid solution hardening ($\sigma_{s}$), precipitation hardening ($\sigma_{p}$), and dislocation hardening ($\sigma_{d}$). The yield strength $\sigma_{0.2}$ can be expressed by the following equation [49]:(8)$$\sigma_{0.2}={\;\sigma }_{g}+\sigma_{s}+{\;\sigma }_{p}+\sigma_{d}$$

For cast alloys, dislocation strengthening can be ignored [50]. Grain boundary hardening is generally estimated by the following formula [51]:(9)$$\sigma_{g}={\;\sigma }_{0}+kD^{-1/2}$$ where k is the Hall–Petch slope, D is the mean grain size, and $\sigma_{0}$ is the lattice friction stress. In this study, the parameter value of *k* = 7.9 MPa$\sqrt{mm}$ [52]. Thus, $kD^{-1/2}$ values of the alloys were calculated to be 17.1 MPa and 16.8 MPa for the Ti70 and Ti80 alloys, respectively. The expression of the lattice friction stress is as follows [53]
(10)$$\sigma_{0}=\frac{2G}{1-\nu }exp\left(\frac{-2{\pi \omega }_{0}}{b}\right)exp\left(\frac{-2{\pi \omega }_{0}}{bT_{m}}T\right)$$ where G and ν are the shear modulus and Poisson’s ratio of the materials, respectively. $\omega_{0}$ is the dislocation width at 0 K, and T and $T_{m}$ are the testing temperature and melting point, respectively. To calculate the G, the RoM was applied, i.e., $G=\sum c_{i}G_{i}$, where $G_{i}$ is the shear modulus of the *i*th element. The $\sigma_{0}$ for Ti70 (298 K), Ti70 (77 K), Ti80 (298 K), and Ti80 (77 K) was, in order, evaluated to be 123.6 MPa, 255.2 MPa, 128.4 MPa, and 261.9 MPa.

It is widely accepted that solid solution hardening in MPEAs is attributed to the high atomic misfit and modulus misfit [54,55]. The solid solution strengthening effect, $\Delta \sigma_{i}$, originating from the *i*th element can be expressed as follows [18,37,56]:(11)$$\Delta \sigma_{i}=AGf_{i}^{4/3}c_{i}^{2/3}$$ where A is a material-dependent dimensionless constant (here A = 0.04 [57]), and $f_{i}^{\;}$ can be computed via:(12)$$f_{i}=\sqrt{\delta_{G_{i}}^{2}+\alpha^{2}\delta_{r_{i}}^{2}}$$ where $\delta_{G_{i}}=\left(\frac{1}{G}\right)\frac{dG}{dc_{i}}$ represents the shear modulus mismatch and $\delta_{r_{i}}=\left(\frac{1}{r}\right)\frac{dr}{dc_{i}}$ denotes the atomic radius mismatch. α is a constant and depends on the type of mobile dislocations. Here, α was designated to be 9 [56]. Since any *i*-element can neighbor with other elements in MPEAs, the $\delta_{G_{i}}$ and $\delta_{r_{i}}$ can be expressed in the following:(13)$$\delta_{G_{i}}=\frac{9}{8}\sum c_{j}\delta_{G}{}_{ij}$$
(14)$$\delta_{r_{i}}=\frac{9}{8}\sum c_{j}\delta_{r}{}_{ij}$$ where $c_{j}$ is the atomic fraction of jth element. $\delta_{G}{}_{ij}$ describes the difference in shear modulus between elements i and j, which can be expressed as $\delta_{G_{ij}}=2\left(G_{i}-G_{j}\right)/\left(G_{i}+G_{j}\right)$ while $\delta_{r_{ij}}=2\left(r_{i}-r_{j}\right)/\left(r_{i}+r_{j}\right)$ describes the difference in atomic radius between elements i and j. $G_{i}$, $G_{j}$, $r_{i}$, and $r_{j}$ are the shear modulus and atomic radius of elements i and j, respectively. The solid solution strengthening was acquired by summation of $\Delta \sigma_{i}$ of each constituent element through:(15)$$\sigma_{s}={\left(\sum \Delta \sigma_{i}^{3/2}\right)}^{2/3}$$

Basic information of constituent elements is listed in Table 3. Consequently, the values of $\sigma_{s}$ for both Ti70 and Ti80 alloys were 427.8 MPa and 355.7 MPa, respectively.

Since the size and volume fraction of the B2 phase in the Ti70 alloys could not be counted, the precipitation strengthening was not considered here. The lamellar HCP phase in the Ti80 alloys could be calculated as a secondary phase. According to the literature, the strength of a lamellar HCP phase is linearly related to its length dimension [58]. Thus, the HCP phase strength contributing was approximately 120.7 MPa by calculation.

Based on the aforementioned details, the strengthening contribution from the individual mechanism is presented in Figure 7. The difference between the theoretical and experimental values for these alloys is less than 6%, except for Ti70 (77 K). The calculated value was consistently in agreement with the experimental data. The reason for the error of Ti70 alloys at 77 K may be that the existence of the B2 phase affected the slip and interaction of dislocations. At the same time, the strength calculation of B2 require accurate phase volume fraction and size [59]. It is difficult to quantify the contribution of B2 precipitation strengthening. The selection of parameters in the whole formula of the theoretical value also brings a certain calculation error.

## 4. Conclusions

In this work, nonequiatomic Ti_70_Al_15_V_15_ and Ti_80_Al_10_V_10_ lightweight MPEAs were fabricated. The crystal structure, mechanical properties, and strengthening mechanism were systemically investigated. Based on the results and analysis, the following conclusions can be drawn:(1)Ti_70_Al_15_V_15_ and Ti_80_Al_10_V_10_ alloys exhibit different phases and compositions. The former showed a BCC structure with a small amount of B2 phase and the latter with a BCC and HCP dual-phase structure. Both lightweight MPEAs displayed uniform equiaxed grains, but Ti_80_Al_10_V_10_ showed some precipitation of the lamellar HCP phase.(2)Upon tension, the yield strength and plasticity of the Ti_70_Al_15_V_15_ alloys were 536 MPa and 13% at 298 K while the yield strength increased to 895 MPa at 77 K. The Ti_80_Al_10_V_10_ alloys showed a yield strength of 657 MPa and a plasticity of 10% at 298 K, and the yield strength rose to 802 MPa at 77 K. When the deformation temperature decreased from 298 K to 77 K, the increase in critical resolved shear stress of dislocation motion was the main factor of strength improvement.(3)Dislocations were the main deformation mechanism in both lightweight MPEAs. The phase interface density was higher in Ti_80_Al_10_V_10_ with a dual-phase structure compared to Ti_70_Al_15_V_15_. The phase interface formed an obstacle to dislocation movement, resulting in significant work hardening in the Ti_80_Al_10_V_10_.(4)Through the calculation of the strengthened theoretical model, the theoretical yield strength was in good agreement with the experimental one. The dominant strengthening mechanisms were found to be grain boundary hardening and solid solution hardening.

## Figures and Tables

**Figure 1 entropy-24-01777-f001:**
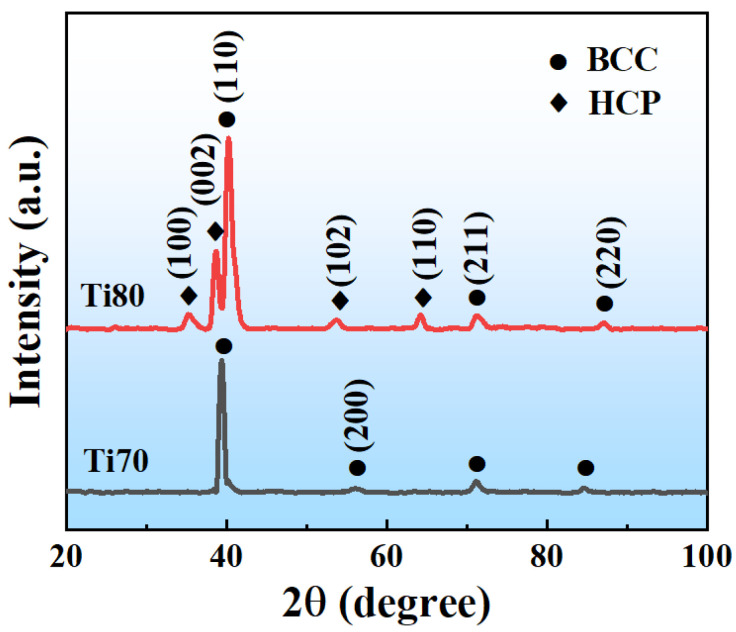
XRD patterns of Ti70 and Ti80 alloys.

**Figure 2 entropy-24-01777-f002:**
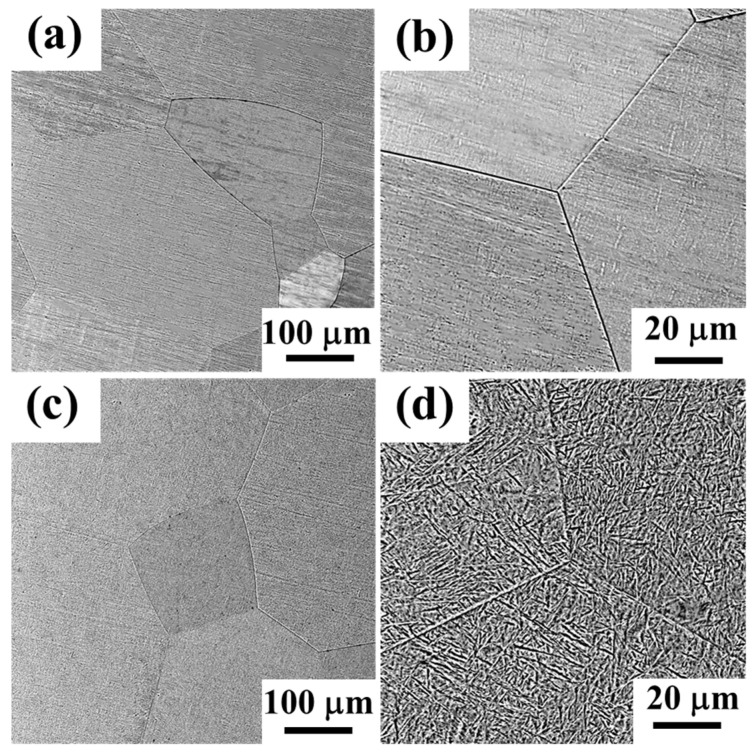
SEM micrographs for Ti70 (**a**,**b**) and Ti80 (**c**,**d**).

**Figure 3 entropy-24-01777-f003:**
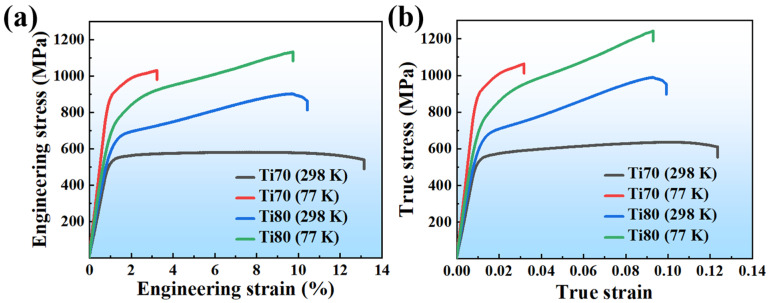
Mechanical behavior of Ti70 and Ti80 alloys at 298 K and 77 K. (**a**) The engineering stress-strain curves and (**b**) the true stress-strain curves.

**Figure 4 entropy-24-01777-f004:**
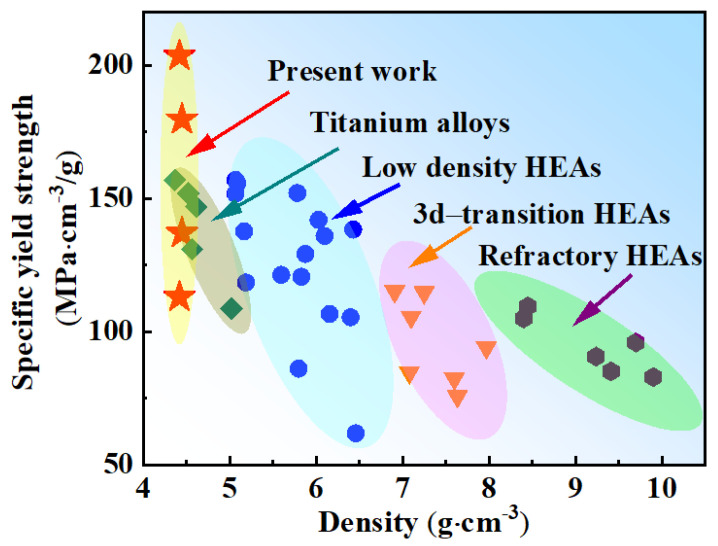
Comparison between specific strength densities of Ti70 and Ti80 alloys with previously reported MPEAs.

**Figure 5 entropy-24-01777-f005:**
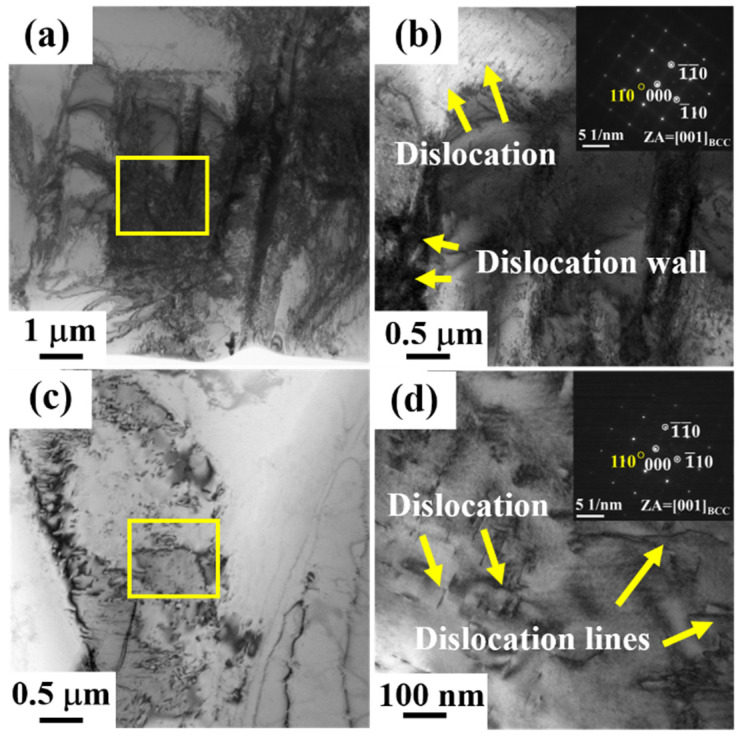
(**a**,**b**) The TEM bright field images and corresponding selected area electron diffraction of Ti70 alloys at 298 K, and (**c**,**d**) the TEM bright field images and corresponding selected area electron diffraction of Ti70 alloys at 77 K.

**Figure 6 entropy-24-01777-f006:**
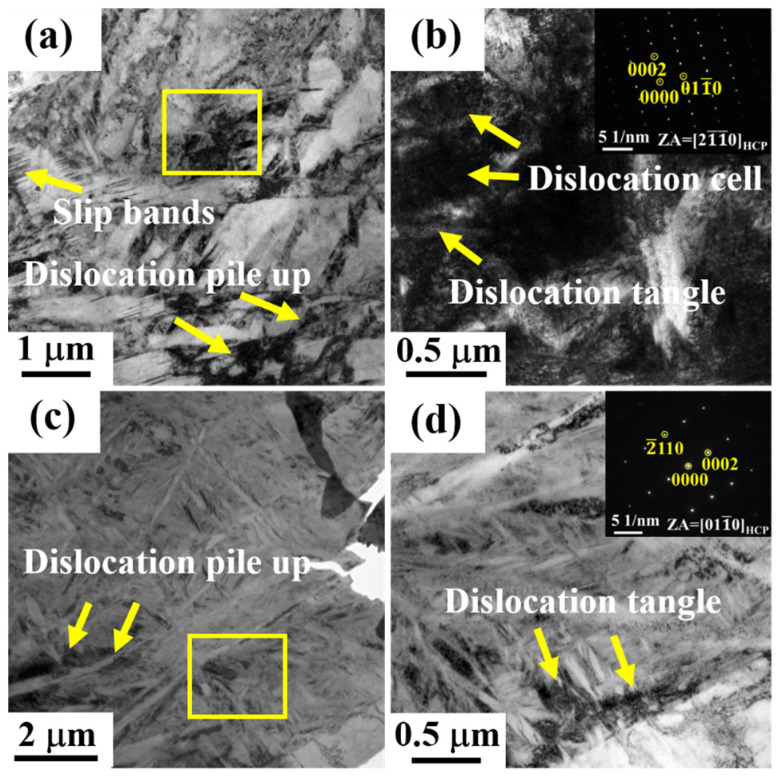
(**a**,**b**) The TEM bright field images and corresponding selected area electron diffraction of Ti80 alloys at 298 K and (**c**,**d**) the TEM bright field images and corresponding selected area electron diffraction of Ti80 alloys at 77 K.

**Figure 7 entropy-24-01777-f007:**
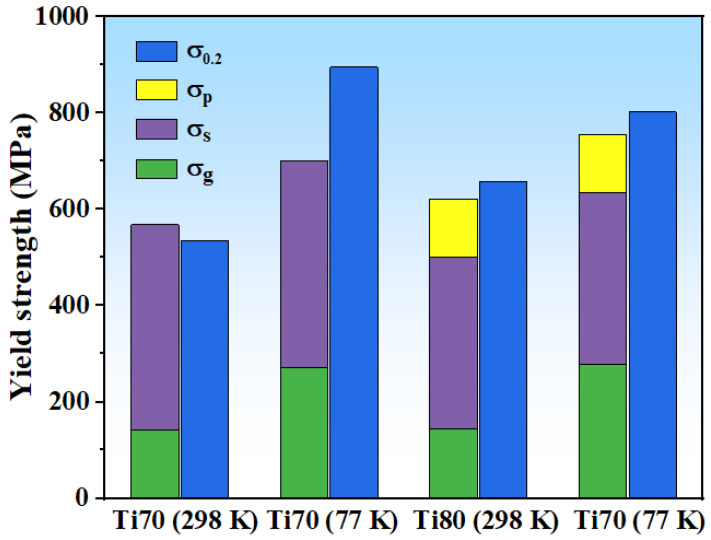
The calculation and experiment yield strength of Ti70 and Ti80 alloys.

**Table 1 entropy-24-01777-t001:** The atomic size difference, δ, enthalpy of mixing, $\Delta H_{mix}$, entropy of mixing, $\Delta S_{mix}$, melting temperature, $T_{m}$, and parameter, Ω, as well as electronegativity difference, Δχ, and valence electron concentration (VEC) of the two MPEAs.

Alloy	δ (%)	$\Delta H_{mix}$ (KJ/mol)	$\Delta S_{mix}$ (J/K mol)	$T_{m}$ (K)	Ω	Δχ (%)	VEC
Ti70	4.10	−14.88	6.81	1815	0.83	4.73	4
Ti80	4.01	−10.88	5.31	1853	0.90	4.94	4

**Table 3 entropy-24-01777-t003:** Density and physical parameters of used elements.

	Ti	Al	V
Density (g/cm^3^)	4.51	2.70	6.11
Atomic radius (pm)	146	143	131
Shear modulus (GPa)	45	25	47

## Data Availability

Not applicable.

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
