# Peer review of "Lightweight Multiprincipal Element Alloys with Excellent Mechanical Properties at Room and Cryogenic Temperatures"

_entropy, 2022, doi:10.3390/e24121777_

Round 1

Reviewer 1 Report

This paper presents a study about multi-principal element alloys with lightweight and excellent mechanical properties. The MPEAs in this study was designed based on the concept of high entropy alloys and achieve low density with well-balanced strength-ductility synergy. The phase of the alloys is main BCC crystal structure. The paper is well-structured and discussed. It provides a method for designing high-performance titanium alloys, which may be of interest to the readers of the field, working in this area. Therefore, I recommend this manuscript for publication but some explanations are needed before it can be accepted for publication in the journal.

1、        Compared with the traditional titanium alloy, such as Ti6Al4V, how about the performance of the alloys in this study. The data of the common titanium alloy should be listed in Figure 4.

2、        There is no B2 phase in the TEM picture and XRD patterns. How to identify the existence of this phase?

Reviewer 2 Report

1. Conditions for the heat treatment of Ti70 and Ti80 alloys are not explained In Materials and Methods section. All the alloys were tested in as-cast state? 

2. In the section of 3.1, the authors said that MPEAs designed in this study meets the empirical parameters for the formation of solid solution. However, Both Ti70 and 80 alloys exhibited BCC + B2 and BCC + HCP phases, respectively. It looks like a contradictory statement.

Reviewer 3 Report

Review report for manuscript entropy-2039567: "Lightweight multi-principal element alloys with excellent 2 mechanical properties at room and cryogenic temperatures" by Lin, Guo, Shi, Han, and Qiao.

Hard materials with high strength and with decent ductility are of high interests in many different applications. If the materials also have low density then the interest is even higher. The manuscript by Lin, Guo, Shi, Han, and Qiao reports promising results from testing of new light but strong materials Ti70Al15V15 and Ti80Al10V10. The manuscript is well written.

However, it would be nice to include a discussion where the results of Ti70Al15V15 and Ti80Al10V10 are compared to the popular Ti6Al4V. It would also be nice to mention a few applications where Ti70Al15V15 or Ti80Al10V10 would be a better choice than Ti6Al4V.

A few corrections needed:

Page 1 – “It has been reported that body-centered-cubic (BCC) MPEAs generally exhibit high strength but limited plasticity, while face-centered-cubic (FCC) MEPAs have relatively low strength but high plasticity.” Reference needed.

Page 1 – “In addition, these MPEAs mainly contain relatively heavy elements, e.g., Co, Cr, Fe, Mn, and Ni for FCC structures, and Hf, Nb, Ta, Mo, and W for BCC structures.” Reference needed.

Page 2 – More information about the XRD measurements are needed.

Page 8 – Please describe “cryogenic deformation” (or should it be “deformation at cryogenic temperatures”?)

Page 8 – Please rephrase “The cross-distributed lamellar HCP phases obstruct each other during plastic strainining and promote local strain concentration, consequently resulting in easier dislocation pile ups and entanglement, thus maintaining excellent strength and plasticity of the alloys [42].”

Figure 4 – Please write “Refractory HEA” instead of RHEA.

Please go through the text and correct all typos, e.g., add space between reference and word, add space between value and unit, remove double spaces, and remove hyphenations where it is not necessary.

Round 2

Reviewer 3 Report

Review report for manuscript entropy-2039567 second draft: "Lightweight multi-principal element alloys with excellent 2 mechanical properties at room and cryogenic temperatures" by Lin, Guo, Shi, Han, and Qiao.

Hard materials with high strength and with decent ductility are of high interests in many different applications. If the materials also have low density then the interest is even higher. The manuscript by Lin, Guo, Shi, Han, and Qiao reports promising results from testing of new light but strong materials Ti70Al15V15 and Ti80Al10V10. The manuscript is well written.

However, a few corrections were needed in the first draft and there were a few suggestions mentioned in the previous review report. Unfortunately has the authors ignored some of them. For example (expressed more clearly):

1. Include a discussion where the results of Ti70Al15V15 and Ti80Al10V10 are compared with the popular Ti6Al4V. Ti6Al4V is a commercial product with very good properties. In what way would Ti70Al15V15 and Ti80Al10V10 be better? (In other words - why should I find these materials interesting?)

2. Provide a few applications where Ti70Al15V15 or Ti80Al10V10 would be a better choice than Ti6Al4V.

3. In Figure 4 – Please write “Refractory HEA” instead of RHEA.

4. Please go through the text and correct all typos, e.g., add space between reference and word, add space between value and unit, add space after comma.

5. Replace SADE with SAED on page 8 on two places ( rows 259 and 265).
